# Trade for Food Security: The Stability of Global Agricultural Trade Networks

**DOI:** 10.3390/foods12020271

**Published:** 2023-01-06

**Authors:** Xiang Wang, Libang Ma, Simin Yan, Xianfei Chen, Anna Growe

**Affiliations:** 1College of Geography and Environmental Science, Northwest Normal University, Lanzhou 730070, China; 2Key Laboratory of Resource Environment and Sustainable Development of Oasis, Lanzhou 730070, China; 3Northwest Institute of Urban-Rural Development and Collaborative Governance, Lanzhou 730070, China; 4Institute of Geography, Faculty of Chemistry and Earth Sciences, Heidelberg University, 69120 Heidelberg, Germany

**Keywords:** economic integration, global agricultural trade, complex network, community division, stability, food security

## Abstract

Global food production is facing increasing uncertainties under climate change and the coronavirus pandemic, provoking challenges and severe concerns to national food security. The role of global agricultural trade in bridging the imbalance between food supply and demand has come to the fore. However, the impact of multifaceted and dynamic factors, such as trade policies, national relations, and epidemics, on the stability of the agricultural trade network (ATN) needs to be better addressed. Quantitatively, this study estimated grouping characteristics and network stability by analyzing the changing global ATN from 1986 to 2018. We found that the evolution of global agricultural trade communities has gone through four stages: the dominance of the US–Asian community, the rise of the European–African community, the formation of tri-pillar communities, and the development of a multipolar community with a more complex structure. Despite witnessing a progressive increase in the nodal stability of the global ATN during the decades, particular gaps can still be found in stability across countries. Specifically, the European community achieved stability of 0.49 and its trade relations were effectively secured. Meanwhile, the remaining leading communities’ stability shows a stable and upward trend, albeit with more significant challenges in trade relations among some of them. Therefore, how to guarantee the stability of trade relations and strengthen the global ATN to resist external shocks has become an essential question to safeguard global food security.

## 1. Introduction

With the deepening development of globalization and economic integration, the economic ties among countries are increasingly strengthened, gradually forming a coexistence of interdependence and competition, as well as mutual penetration and constraints [1,2]. Recently, more and more countries have sought greater profits through global agricultural trade, or in this way, to fill the gap in their supply and establish an international channel for the transfer of agricultural surplus and shortage [3,4]. Hence, agricultural trade plays an essential role in promoting the new wave of economic growth and the transregional allocation of arable land and water resources [5,6,7]. Between 1961 and 1970, global trade in agricultural products increased by USD 18.6 billion, while from 2009 to 2018, it rose by USD 487.8 billion, with a remarkable acceleration in its growth [8]. As the diet structure of developing countries changes, the gap between global supply and demand for agricultural products is expected to increase continuously in the future, and the trade in agricultural products will also grow further [9,10]. Therefore, it is critical to analyze global food security from the perspective of agricultural trade for the discussion of sustainable development worldwide.

Recently, the stability of the agricultural trade network (ATN) has been dramatically challenged by heightened trade frictions and the worldwide spread of the pandemic. On the one hand, vital exporting countries, led by the United States, have strengthened export controls on some major importing countries; on the other hand, other significant exporters, such as Russia and Ukraine, are affected by the epidemic and their national relations which impede their contribution in international markets [11]. Along with the rapid growth of the global South, unilateralism and trade protectionism have surfaced, and the increasing trade frictions and escalating conflicts are causing a sharp upsurge in potential risks to the ATN [12]. Since 2020, key international wheat exporters—Kazakhstan and Russia—and the leading rice exporter—Vietnam—have enacted a series of protective export bans, which has again shaken the global trading market. From a global perspective, the stability of the ATN is conducive to ensuring efficient and robust operations across the agricultural system. However, the US–China and Russia–Ukraine conflicts present not only a local impact on food security but also a noticeable shock on a global scale. Agricultural trade is pivotal for those countries with insufficient domestic resource supplies for individual countries, especially for populous and rapidly growing economies such as China [13]. Trade stability is essential for regional and global food security, economic prosperity, and sustainable development [14]. It is economically beneficial, not only for exporting countries but also for importing countries, to receive sufficient food. The entire trade network is expected to be at risk if the current dynamic and volatile trade pattern further intensifies, especially by the breakdown of relations between the core countries. Therefore, dealing with potential risks and integrating more profoundly but steadily into the global agricultural economy has become an urgent issue that must be addressed to eliminate hunger and achieve sustainable food development.

Complex network analysis has become an effective method for understanding trade characteristics. The general idea is to construct the corresponding networks by taking the countries involved in the trade as nodes and the connecting lines between two nodes as edges [15]. Some studies have built trade networks for oil [16,17], natural gas [18], and advanced manufacturing [19] to reveal their trade characteristics and evolution. With the rapid development of global agricultural trade, its related network studies have also grown [20,21,22], focusing on topics such as the construction of a global ATN of major agricultural products and the analysis of their evolutionary characteristics [23]. In addition, researchers have also investigated the features of resources embodied in agricultural trade from the perspective of arable land and water resource flows [24,25,26,27,28]. Major producing countries, represented by countries such as Brazil, were found to be key players in meeting the growing global food demand and influence the stability of the ATN significantly [29]. With the complexity of the global agricultural trade environment, trade stability emerged as a new research hotspot. However, existing studies are limited to qualitative methods to analyze specific objectives, such as cereals, fruits, and fishery products, or quantitative ones, through individual indicators such as geographical concentration [30]. Since the stability of agricultural trade is affected by multiple factors, such as political risks, economic benefits, geographical distance, national relations, and cultural attributes, its corresponding research needs to be conducted throughout the network.

The global ATN is a complex system with multiple nodes and links. It encompasses many countries and products and implies intense vulnerability and low-risk resistance. Such network attributes, thus, also necessitate understanding individual country-related trade risks or opportunities based on the analysis of the stability characteristics for the entire network. The paper is structured as follows: in the introduction, an overview of the importance of ATN stability for global food security is presented, along with a review of existing literature and its limitation. Subsequently, the methodology (complex network analysis), including the steps of community division and ATN stability calculation, is presented. The results analysis section shows the changes in the community structure and the evolution of the stability of the ATN and the main nodes. Finally, in the discussion section, we explore ATN stability’s impacts and policy implications for global food security, identify current barriers and challenges, and discuss potential solutions.

In particular, this paper attempts to answer the following questions:(1)What is the dynamic evolution of the structure of the global ATN community?(2)Which changes characterize the stability of the global ATN and how do changes in key nodes affect the overall stability of the network?(3)Can the stability of major importing countries be guaranteed at present? How great is the potential threat to global food supply security from disruptions in major trade relations?

## 2. Method and Materials

### 2.1. Methodology

According to complex network theory, the ATN can be represented by the set N = (P, E), where P denotes a country involved in agricultural trade, i.e., a node in the network, P = {P1, P2, ..., P228} (the total number of 228 here is obtained on the basis of the number of countries trading in the FAO database, excluding some island countries and regions). E denotes the line between two nodes, i.e., the edge of the network, thus building the ATN [31,32,33].

#### 2.1.1. Regional Characteristics of Global Agricultural Trade Networks

Due to the limitations in the attributes of agricultural products, the risks of geopolitics, and the similarity in trade preferences among some countries, agricultural trade shows strong grouping characteristics. We thus employ community division, with its structural characteristics, for our analysis.

##### Community division and structural characteristics

A community in a complex network is a collection of closely connected nodes and represents an important indicator of the structural characteristics within the network area [34]. A community has strong trade dependencies, while the connections between two communities are relatively sparse. Newman calculated communities by the concept of “modularity”, which refers to the proportion of edges connecting the internal vertices of the community structure in the network minus the expected value of the proportion connecting two randomized nodes in an equivalent network; the number of edges is measured by the quantitative function *Q* [35]. In a directed network, the mathematical formulation is provided below [36]:(1)$$Q=\frac{1}{m}\sum_{\mathrm{ij}}[a_{\mathrm{ij}}-\frac{k_{i}k_{j}}{m}]\delta (c_{i},c_{j})$$
where $m=\sum_{\mathrm{ij}}a_{\mathrm{ij}}$ denotes the sum of the weights in the network, $c_{i}$ represents the community to which node i belongs. Here, *a*_ij_ is total trade volume between node i and node j; *k*_i_ and *k*_j_ denote the trade volume of node i and node j, respectively. If node i and node j are in the same community, $\delta (c_{i},c_{j})=1$, and if they are not in the same community, it is 0. The modularity formula simplifies the situation as follows:(2)$$Q=\sum_{c}[\frac{\sum_{\mathrm{in}}}{m}-{\left(\frac{\sum_{\mathrm{tot}}}{m}\right)}^{2}]$$
where $\sum_{\mathrm{in}}$ denotes the weight value inside community c and $\sum_{\mathrm{tot}}$ indicates the weight of the edges connected to the points inside community c, and includes all edges inside and outside the community.

The following steps are employed in this research to achieve an appropriate identification and classification of the community structure [37]:Start by considering each node in the trade network as an independent community.Divide the community $c_{i}$ and $c_{j}$ into a new community and calculate the variation of the function value $\Delta Q_{\mathrm{ij}}$.Then, select the largest $\Delta Q_{\mathrm{ij}}$, and divide the corresponding two communities into a new one.Repeat steps (i) and (ii) until the community structure cannot be further divided.

##### Level of trade satisfaction

The level of a community to satisfy the demand of its internal countries determines the stability of the ongoing trade. Moreover, when the country has difficulties meeting the current community’s import demand, it will switch to other communities to secure a stable supply of agricultural products. Therefore, the trade satisfaction index is proposed, expressed as:(3)$$S_{i}=\frac{k_{i}^{\mathrm{com}}}{k_{i}^{\mathrm{in}}}$$
where $S_{i}$ denotes the degree of trade satisfaction that country i receives from its community, $k_{i}^{\mathrm{com}}$ refers to the total amount of agricultural products imported from its community, and $k_{i}^{\mathrm{in}}$ represents the total amount of imports into country i. The larger the $S_{i}$ is, the less country i is vulnerable to trade supply shocks from other communities and the more concentrated the source of imports is [16].

#### 2.1.2. The Stability of the Global Agricultural Trade Network

##### Node stability

Trade policies, national relations, epidemics, wars, and many other factors can cause disruptions in the trade of exporting countries in the network. When there is a complete disruption of external agricultural supplies within the network, we calculate the ability of the importing country to seek new sources as:(4)$$CV(j)=\sum_{i=1}^{v_{N_{\mathrm{ex}}}}P_{i}\times \frac{X_{i}}{X_{w}}\times d_{\mathrm{ij}}^{-1}$$
where CV(j) denotes the stability of agricultural importing countries and $N_{\mathrm{ex}}$ is the number of exporting countries in the network. $P_{i}$ represents the exporting country’s political security index; corresponding data can be obtained from the Country Risk Guide database. A higher value indicates a stronger ability to guarantee steady supplies. $X_{i}/X_{w}$ denotes the share of country i’s exports in total global exports. $d_{\mathrm{ij}}$ indicates the distance between two national capitals, obtained from the French Geographic Database. Once subject to trade disruption, the stability of the agricultural importing country is determined by the security, accessibility, and promptness of obtaining new suppliers, corresponding to the three variables in equation (4): *P*, *X*, *d*. Here, larger nodal stability indicates a greater ability of the agricultural importing country to obtain a fast and stable supply [16].

##### Network stability

The stability of the agricultural trade network refers to the ability of others in the network to obtain rapid supply if the “hub” node in the network is threatened or attacked from outside. The stability of the network can be expressed as:(5)$$NV=\sum_{j=1}^{N_{\mathrm{im}}}\frac{M_{j}}{M_{w}}\times CV(j)$$
where NV denotes the stability of the network and $M_{j}/M_{w}$ shows the share of agricultural imports of country j in global imports. In general, the more stable the network is, the stronger the ability of the network to cope with risks and the less likely it is to “fail” [16].

### 2.2. Data Sources

In this study, data were obtained from the global bilateral trade statistics in agricultural products published by the Food and Agriculture Organization (FAO) of the United Nations for the period 1986–2018 [9]. We examined the spatial–temporal evolution and stability of the trade network comprising 228 countries. Considering the inter-annual variation in purchasing power due to inflation, we discounted the trade volume for all 33 years at constant 2020 prices [38]. In particular, 87 agricultural products in 10 major categories are included in this study (Table 1).

## 3. Results

### 3.1. Characteristics of Global Agricultural Trade Network Communities from 1986 to 2018

#### 3.1.1. Change of Community Structure

According to the characteristics of the changing structure of communities in the global ATN, four stages are outlined (Figure 1):(1)Period of the dominance of the American–Asian community (1986–1997)

In 1986, the global ATN was mainly divided into three leading communities, among which the East Asia–America community was in the absolute core position (Figure 1a). The European–African community ranked second, with France and Italy as the trade hub. In the case of the East Europe–Asia community, which included 18 neighboring countries led by the former Soviet Union, its trade volume accounted for only 16% of the world. The overall structure of the communities in this period was characterized by the dominance of the East Asian–American community. The local network structure was centered on the United States, while China, with its small trade volume, was only a peripheral country by then. This core–periphery structure clearly distinguished the strengths and weaknesses of the agricultural economic system during the period.

(2)Period of the prompt rise of the European–African community (1997–2008)

Due to the expansion of trade caused by the deepening of global integration, the period witnessed the rapid rise of the European–African community (Figure 1b). Along with an increase in its number of countries and trading volume, it gradually occupied half of the ATN and became an essential part of the global food system. At the same time, the former Eastern Europe–Asia community slowly disintegrated, and a new Asia–South America community was developed. As for the Middle East, which was active in the American–Asian community before the 1990s, it has gradually left its former community and gravitated towards economic ties with emerging agricultural exporters, such as Brazil and Argentina, due to the growing tensions with the United States. It is evident that the influence of state relations on agricultural economic linkages is exacerbated. The increased attractiveness of some core agricultural countries facilitated the emergence of new communities.

(3)Phase of gradual formation of tri-pillar communities (2008–2018)

After entering the 21st century, the rapidly developing Asian economies and the rising income levels of their residents have led to expanding demand for high-value agricultural products. During this period, China gradually broke away from the American community into the Asian one, indicating the end of a completely American-dependent agricultural trade in China (Figure 1c). The economic crisis that began in 2007 had a considerable impact on the European and American markets, resulting in many countries, especially those in Africa, switching to the Asian community and contributing to its development. The tri-pillar pattern was formed in the global ATN, led by the Asian–South American community (with Brazil and China as the core), the American community (with the US and Japan as the core), and the European community (with Germany and France as the core).

(4)The development of a multipolar community with a more complex structure (2018 to present)

Nowadays, the global ATN is generally characterized by “diversification” and “stabilization”. The Central Asia–North Africa community has become a strong force in this network by strengthening its trade connections (Figure 1d). Meanwhile, benefiting from the favorable trade environment and the interoperability of agricultural trade, the trade volume of this community is expected to grow further in the future. Under the influence of the trade friction between the US and China in 2017, China adjusted its trade strategy and expanded imports from Brazil, Argentina, and countries along the Belt and Road (e.g., Russia and Kazakhstan). It strengthened the internal ties of the Asia–South America community and transcommunity ties with Central Asia and North Africa. Forming a multicore trade pattern is conducive to a more sustainable and stable development of the global food system.

The structural evolution of global ATN communities was coupled with the pattern of global population increase from 1986 to 2018. Africa and the Middle East were the two “hot spots” of population growth, and such a rapid increase has brought about continuous rises in demand for agricultural products. It also promoted the emergence of Central Asia–North Africa and South Africa communities at this stage, making the division of trade communities more diversified. Simultaneously, the world’s leading economies have brought their resource endowments and comparative advantages in production into the agricultural trade, thus becoming the hub nodes in their respective communities and playing dominant roles in the local trade networks. Since the 21st century, the growing population and rising income levels in the global South, in countries such as China, have led to a rapid increase in food consumption demand, prompting its integration into the wave of economic globalization. Notably, although China is already the largest importer in the global agricultural trade network, it is still exposed to significant trade risks and instability.

#### 3.1.2. The Satisfaction of Trade in Major Communities

Communities in the trade network provide strong import security for the countries within them. In case of a high degree of trade satisfaction, the agricultural imports would be less affected by other communities, and a subsystem of a global ATN would be formed within the community. We selected the top 10 countries for analysis based on the magnitude of trade satisfaction at each stage, and the results are shown in Figure 2. It is particularly relevant for those countries with high external dependence, such as Japan, which is perennially relying on its community core, the United States. The United States has a robust agricultural resource endowment, which allowed Japan to obtain a demand satisfaction level above 90% in all our observation years (1986, 1997, 2008, and 2018, see Figure 2). However, if the trade concentrates excessively in the core countries inside the community, once trade friction occurs or related trade policy tightens, their agricultural trade links would be interrupted, and the domestic supply would face a considerable threat. Therefore, although such a trade model poses fewer threats from other communities, it is considered to remain potentially risky. In addition to Japan, Germany and Mexico are high-value trade satisfaction countries. In comparison, China, the Netherlands, and Belgium maintained a relatively lower satisfaction level, in the range of 50–70% during 2008–2018, which also implies a dispersal of import risks. This reasonable risk aversion enables more secure agricultural imports into these countries.

### 3.2. Stability of Agricultural Product Trade Network

#### 3.2.1. Node Stability Changes

The node stability of the trade network is in a dynamic process of change in space and time. During 1986 to 2018, the average value of the node stability of the global ATN increased from 0.24 to 0.39; despite an increase of 0.15 over 33 years, the overall stability still needs to be improved. Benefiting from the improved accessibility among countries within the trade network, they actively expand their sourcing countries to ensure import stability. During this period, there is a clear hierarchical structure in node stability of the global ATN (Figure 3). The top five node stability rankings in research years all belong to EU countries, mainly attributed to the proximity of trade transportation and their stable trade relations and political environment. As for the importing countries, such as China, they have continued to increase during the study period, which indicates strong protection of agricultural import security there. Overall, with the development of global agricultural trade, the global ATN has witnessed increasing steadiness of various nodes. It also provides a stronger guarantee for the steady and orderly operation of the whole network. However, the difference across countries further widens in parallel with the increase in node stability, manifested by the rapid improvement for the critical nodes and the insignificant change for the edge nodes in the network. The result shows that in 1986, the highest node stability was 0.69 in Belgium and the lowest stability was only 0.01 in Guinea, while in 2018, this indicator had risen to 0.83 in Belgium but Guinea’s remained at 0.03. Hence, the node stability changes of the global ATN are characterized by the continuous stability of those “head” nodes and the slow changes of the “tail” nodes, leading to an ever-widening gap between them. It is necessary to ensure the stability of the “head nodes” and increase the average level of the “tail nodes” to maintain the overall stability of the global ATN.

#### 3.2.2. Stability of Agricultural Trade Networks in Major Countries

By analyzing the stability of the top five importing countries in the global ATN at the current phase, it is found that the decline in stability is characterized by a non-uniform rate of trade disruptions under varying scenarios (Figure 4a). In particular, the United States has a slower overall decline due to its long-established favorable trade relations and proximity to significant import sources such as Canada and Mexico. Germany has the highest overall stability because most of its import sources are within the EU countries and, therefore, its trade is more accessible, timely, and secure. Japan has adopted a relatively focused import-sourcing policy, with most of its agricultural trade originating from the United States, resulting in the lowest stability. With such a highly concentrated trade policy, which requires friendly and solid relations between trading states, a stable external environment, and a year-round strong export capacity in exporting countries, it is risky in the current context of climate change and globalization transformation.

At a time of trade frictions or trade disruptions, import stability is crucial for China, the world’s largest importer of agricultural products. By interrupting the top 10 source countries of China’s imports (Table 2), findings show that, although China’s agricultural trade has gradually expanded to Brazil, Argentina, and other countries along the “Belt and Road”, such as Russia and Kazakhstan, in recent years, the United States is still an essential source for China by virtue of its stable output level and high export security. In addition, compared with other major importers such as Europe and the United States, China’s stability curve decreases faster after trade disruptions (Figure 4a). It indicates that as the number of disrupted countries increases continuously, the stability of China’s agricultural imports decreases rapidly, and it reacts more sensitively to those changes.

The random sampling process was repeated 100 times for the corresponding settings using Matlab software, and the random interruption mean values were derived (Figure 4b). The mean value of random disruptions is shown to decrease less than the target disruptions, suggesting a more significant impact of target disruptions on the stability of China’s agricultural imports. When one country’s trade is interrupted, China may balance the supply by increasing the trade volume of other countries. However, when supplies from several major source countries break down simultaneously, the security of China’s agricultural imports will be tremendously challenged. Therefore, to maintain the high stability of a giant agricultural importing country like China, it is necessary to stabilize the existing trade relations through long-term international agricultural cooperation and diplomatic efforts. Meanwhile, it is necessary to broaden trade channels and strengthen trade ties with dominant exporting countries.

### 3.3. Changes in Trade Stability by Different Communities

The “head nodes” with high stability are often in the same community, and their long-term trade relationships are likely to further the growth of the stability of the whole community and individual nodes (Figure 5). With the evolution of community relations, the stability of each community also changed. In 1986, the stability of the East Asia–America community was 0.31, with more than 76% of the nodes within the community showing stability greater than 0.2, and only a few marginal nodes, such as Algeria and Nigeria, showing stability lower than 0.05, indicating stable trade relations within this community by then. At that time, the stability of the Europe–Africa community was 0.22, with clearly distinguished stabilities, i.e., the stability of European countries, represented by Belgium, being much higher than that of African countries inside. Eastern Europe–Asia stability is also 0.22, and is low overall, except for the core countries.

In contrast, by 2018, the stability of the European community had increased to 0.49, with stronger intra-community ties and enhanced abilities to cope with external supply disruptions. As for the Asia–South America community, its stability also rose to 0.34, mainly attributable to the boost of the three core countries (China, Argentina, and Brazil), whose stability exceeds 0.7. This community is expected to ensure the security of agricultural supplies in countries of the global South. Additionally, the stability changes of the Southeast Asia–America community has steadily increased from 0.31 to 0.41 over the past 34 years, while the role of the core countries in securing stability within the community has weakened. The emerging Central Asia–North Africa community shows stability of 0.31, with the stability of core countries (Ukraine, Russia, and Egypt) exceeding 0.6. In South Africa, the stability of the community is 0.3. Due to the relatively small number of participating countries, the South African community currently has less impact on the stability of the global network. The stability of each community shows an increasing trend, but the growth rate varies among them. The group rise of developing countries, led by China and Brazil, in the global ATN, has promoted the stability of their respective communities. The United States, as the world’s largest exporter of agricultural products, is an essential component to guarantee the stability of the network globally. Therefore, it is necessary for the US to enhance its export capacity, and further stabilize the existing trade relations and secure global food imports with favorable and consistent trade objectives. Furthermore, as Brazil plays a key role in providing food for the growing global demand, its position in securing global ATN stability cannot be neglected.

## 4. Discussion and Policy Implications

Complex network theory is an efficient way to analyze the structure of trade networks and their evolutionary process. The analysis of the ATN’s community structural changes and stability reveals the clustering characteristics of the agricultural trade and its ability to cope with risks. With the increasing complexity of the global ATN, communities in the network reflecting the core relationships of nodes in the trade network are gradually increasing. The emergence of new communities indicates that the dynamics of agricultural production and demand patterns, state relations, and trade policies have changed the relationships within the original community. Meanwhile, some of the enhanced trade across communities is also attributable to reduced transport costs and improved storage techniques for agricultural products.

The evolution of the global ATN reflects the increasing dependence of more countries on foreign supplies of agricultural products, indicating that with economic globalization, agricultural resources are being reallocated on broader spatial scales with economic globalization, agricultural trade is becoming more interconnected, and the countries involved in the trade network are more closely related to each other. The globalization of agriculture is increasingly important for food security, and a well-functioning international trade network can support sustainable development and climate change-related challenges worldwide [39]. Some of the less productive regions are vulnerable to the effects of climate change, and a free flow of global ATN will guarantee food supplies to these regions, with trade in agricultural products tying the global food system together. Therefore, the stability of the global ATN has important implications for the continued global food supply, especially for countries with high external dependency, represented by Japan and Korea, where the stability of the global ATN will directly impact their supply. In addition, as a result of unexpected factors, concentrated trade relations may increase the sensitivity of importers to price spikes if food production is disrupted or the government restricts exports. From this perspective, agricultural networks are evolving toward a “robust but fragile” structure [40]. Therefore, improving the stability of the ATN, both for import and export, would benefit them; i.e., exporting countries could gain greater economic returns. In contrast, importing countries would secure their supply through trade. The relative paths to realize this improvement, such as managing connectivity within the food system and the incremental changes that affect the food system, also need to be addressed in future studies [41].

Numerous factors affect agricultural products’ trade, particularly the geopolitical environment and long-term trade relationships. The long-established trade relations within the EU have assisted the community in withstanding external shocks. With the CAP implementation, the European community has become the most stable trade relationship worldwide. Overall, in the face of the current trend of shifting the focus of agricultural trade to US–Asian communities, it is a new challenge to establish stable and long-lasting ties with other communities. For example, long-distance vegetable and fruit trade between the European Community and China and trade in livestock products with the Americas can be important ways to address the current supply shortage within the EU community. Establishing such cross-continental trade relations, in turn, requires joint efforts and logistical (e.g., shipping lanes) access on both sides. Therefore, the gradual establishment of global routes and the increase in maritime and air transport capacity contributed to the evolution of the global ATN community structure and ensured the stability of the global ATN.

As the largest importer in the global agricultural trade network and a major exporter of fruits and vegetables, China’s fundamental circumstances of abundant population and scarce water resources make it inevitable that the country must fulfill domestic demand for agricultural products through the international market [42]. However, as the largest importer of global agricultural trade, China’s domestic agricultural supply significantly impacts both its own national and global food security. Therefore, China, as well as those African countries with similar problems (e.g., Egypt, Algeria), should diversify their trading partners, deepen their ties with various nodes and communities in the ATN, and strengthen agricultural cooperation and exchange activities to ensure a stable and secure food supply. Additionally, in response to the high risk of imports from some countries, on the one hand, there is a need to expand the source countries of imports and strengthen cooperation with leading agricultural producers. On the other hand, production techniques should also be improved to enhance agricultural water use efficiency, reduce the loss of high-quality arable land, and gradually increase the overall food production capacity.

As more and more countries rely on global trade to supply their agricultural products, policy openness is likewise gaining significance. Policy changes and economic developments are critical drivers of structural transformation in global agricultural trade [25]. In economic globalization, the breakdown of local trade relations is expected to negatively impact the stability of global trade, and no country involved will be able to remain unaffected. Agricultural globalization is increasingly vital for national food security, and policymakers need to turn away from a regional perspective to a network-oriented and interactive one. Numerous factors impede the stability of the global ATN. For example, it is not only the Chinese food supply that was directly affected by the US–China trade friction in 2017, but also the income of US farms. Moreover, the Russia–Ukraine war since 2022 has blocked both wheat exports from Ukraine and grain exports from Russia, thus influencing the relative supply in various nations. These conflicts have caused food shortages and other crises which have spread to the Middle East, North Africa, South Asia, and even South America. These crises may cause hunger revolutions in many countries, such as Lebanon, Sri Lanka, Egypt, Sudan, and Tunisia. The potential detrimental impact on global markets of export restrictions arising from national conflicts must therefore be carefully weighed against the fact that export restrictions will inevitably exacerbate upward price pressures on international markets and worsen the global situation. In order to respond to such food export restrictions arising from conflicts in major countries, there is a need to strengthen the structure of the global pluralistic network and the stability of leading countries of the community, especially for the marginal countries in the global ATN. COVID-19, on the other hand, affects the global food system in many respects, particularly regarding production disruptions, trade limitations, and economic downturns. Many developing countries have been severely affected by COVID-19 due to export bans issued by 14 major countries, which are detrimental to the operation and stability of the global ATN. The need for a timely and coordinated response could minimize the challenge to global food security. Therefore, win–win cooperation should be the criterion for global agricultural trade. The elimination of trade frictions and barriers should be promoted to jointly address the potential food crisis caused by national conflicts and the COVID-19 outbreak and effectively maintain local and global food security.

## 5. Conclusions

The study focuses on the stability of the global ATN in the context of increasing trade frictions and frequent export bans due to the pandemic. In summary, the three questions introduced can be answered as follows:

(1) The analysis of the division and structural characteristics of the global ATN reveals a four-stage evaluating process: the dominance of the US–Asian community, the rise of the European–African community, the formation of tri-pillar communities, and the development of a multipolar community with a more complex structure. Currently, there is a gradually forming, diversified pattern of coexisting leading communities. The United States, the world’s largest exporter, is at the heart of the Southeast Asia–America community. The stability of its trade is essential to safeguard the core importers within the communities, represented by Japan. As the world’s largest importer, China has gradually transformed itself from a North American–Southeast Asian community to a South American–East Asian community, becoming one of the core countries of the community. The main reason for the change in the structure of the global ATN community is dynamic trade relations. Benefiting from improved transport preservation, storage technology, and the rapid growth of some long-distance trade, it accelerated the emergence of new communities, especially in some localized countries where the volume of trade has increased and new community core countries have been formed.

(2) The stability of the global ATN has increased from 0.24 in 1986 to 0.39 in 2018. The increase of 0.15 in the stability of the global ATN over 33 years indicates that the global ATN moved in a stable direction. The improvement in the stability of the global ATN is guaranteed by the fact that the stability of the main nodes of the global ATN gradually increases over the four phases, especially in the EU countries, represented by Belgium. However, in some marginal countries, stability has remained low, affecting the growth of global ATN stability. Differences in stability among countries widened continuously, with the European community having the highest stability, followed by the Asian–South American community and the Southeast Asia–Central and North American communities. As a giant importer, China’s stability has risen incrementally, from 0.51 in 1986 to 0.69 in 2018, indicating its quest for long-term and reliable sources based on stable trade relations, which is significant to guarantee the stability of the agricultural import side within the community and, even, to ensure global food security.

(3) Taking China as an example, based on our analysis of import stability in the scenario of trade disruptions, we found that China’s stability can still be maintained at 0.69 under the scenario of trade disruptions in its top 10 source countries. However, the stability of China’s agricultural imports will decrease rapidly with a continuous increase in disrupted countries. Since the threat of targeted disruptions is larger than random disruptions, the security of China’s agricultural imports will be severely challenged when several major source countries disrupt their supply simultaneously. Although the sensitivity towards disruptions varies, the declining trend will not change. According to the results of Section 3.1.2, countries such as Japan and Mexico, where trade is concentrated, will also exhibit the same characteristics as China. These countries require long-term stable trade relationships with their communities to ensure food supply security.

There are limitations in this study; for instance, despite using bilateral trade data, our focus is more on the stability of the import side in terms of the sustainability of global food sources. Moreover, the leading importing countries, represented by China, are also chosen for one of our main study objectives.

## Figures and Tables

**Figure 1 foods-12-00271-f001:**
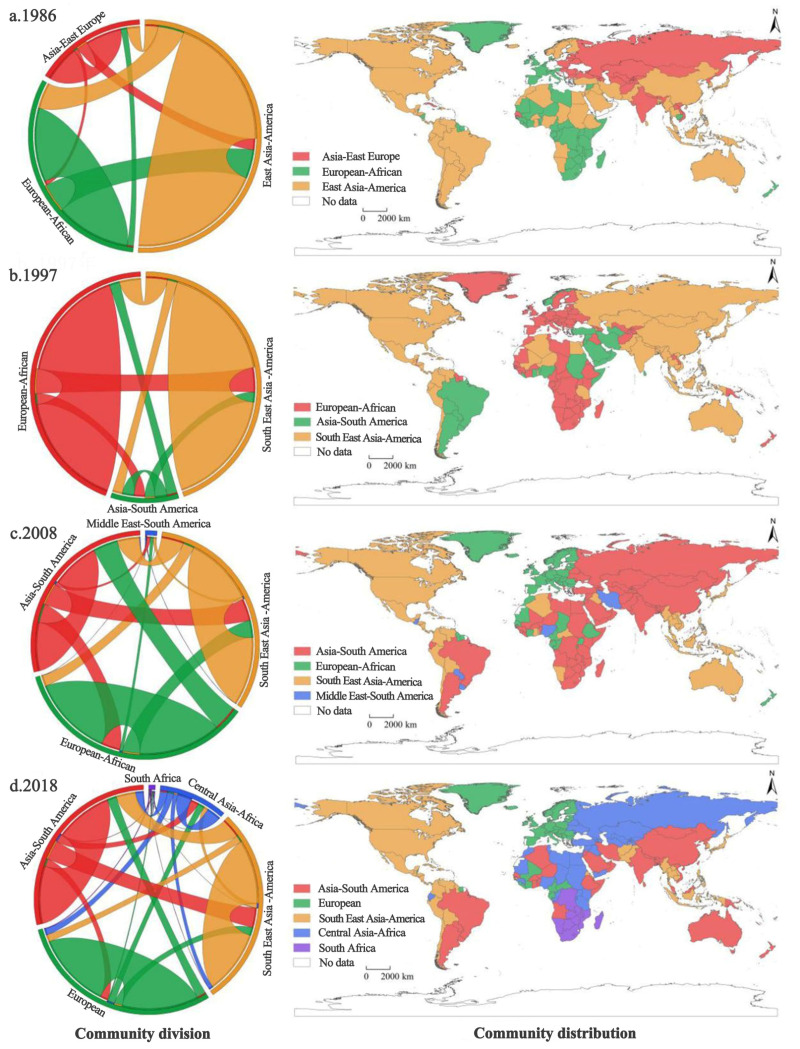
Community structure and distribution of global agricultural products trade network in (**a**) 1986, (**b**) 1997, (**c**) 2008, and (**d**) 2018.

**Figure 2 foods-12-00271-f002:**
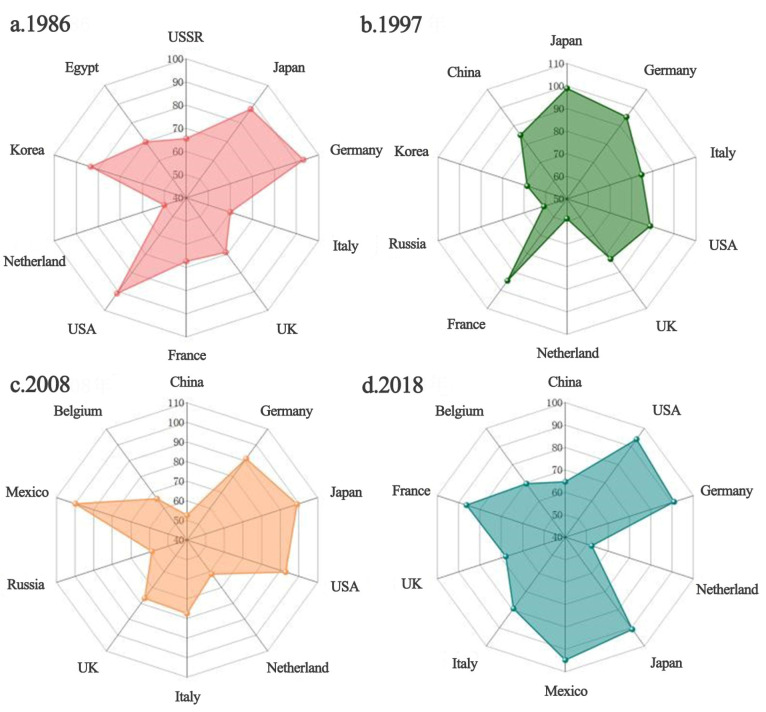
Satisfaction level of global agricultural trade network community in (**a**) 1986, (**b**) 1997, (**c**) 2008, and (**d**) 2018.

**Figure 3 foods-12-00271-f003:**
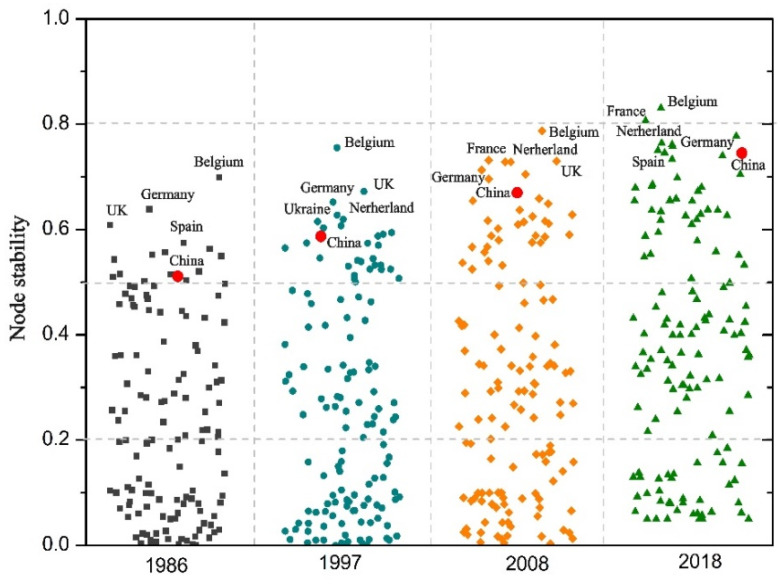
The nodes’ stability changes in the global ATN in the four stages.

**Figure 4 foods-12-00271-f004:**
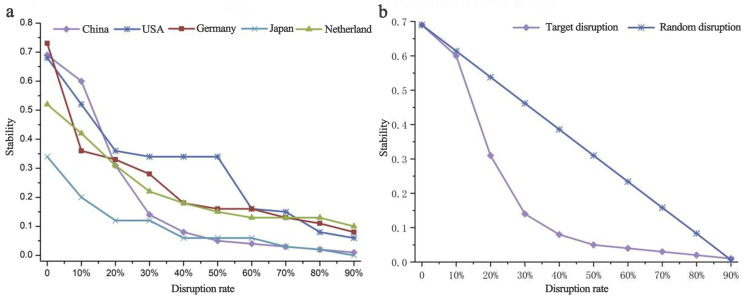
The stability changes in the top five countries (**a**) and China (**b**) under different disruption scenarios in 2018.

**Figure 5 foods-12-00271-f005:**
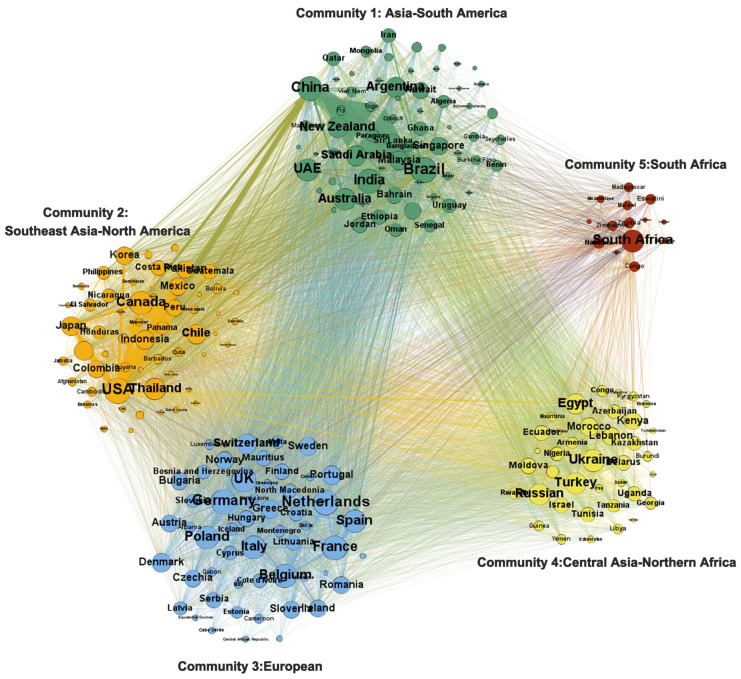
Trade stability of different communities in 2018.

**Table 1 foods-12-00271-t001:** Ten global agricultural trade categories covered in this study.

Categories	Name of Agricultural Products
Cereals	Wheat, maize, rice, millet, barley, buckwheat, sorghum, oats, rye, other grains
Oil	Soybean, peanut, sunflower, flaxseed, rapeseed, sesame, nuts, olive
Fiber	Lint, jute, sisal, ramie, cotton lint, cotton seed, cotton seed oil, cotton seed cake
Sugar	Raw sugar, refined sugar
Potato	Cassava, potato, sweet potato, yam, roots, tubers
Fruit	Apples, pineapples, strawberries, peaches, watermelons, bananas, cherries, pears, mangoes, kiwis, papayas, lemons and limes, grapes, apricots, avocados, blueberries, figs, oranges, plums
Vegetable	Garlic, onions, tomatoes, carrots and radishes, cauliflower and broccoli, cucumbers and gherkins, fennel celery, ginger, cabbage, peppers, bamboo shoots, lettuce and chicory, eggplant
Meat	Pig, cattle, chicken, duck, goat, horse, rabbit, sheep
Milk and eggs	Fresh milk, fresh goat milk, dairy products, eggs
Others	Tea, beer, wine, beverages, coffee, cocoa, pepper, sweet pepper, cloves, other spices

Note: Table 1 classification uses FAO taxonomy standards and corresponds to HS coordination codes.

**Table 2 foods-12-00271-t002:** Stability of China’s agricultural imports in 2018.

Source Countries of China’s Imports	Political Risk Index	Export Capacity Level	Inverse of Shortest Route	Degree of Export Security	Supply Country Interruption Ratio	Stability of China’s Agricultural Imports
Brazil	0.14	0.92	0.07	0.09	0	0.69
United States	0.87	1.00	0.33	0.29	10%	0.60
Canada	0.96	0.35	0.51	0.17	20%	0.31
Australia	0.88	0.22	0.33	0.06	30%	0.14
New Zealand	1.00	0.10	0.29	0.03	40%	0.08
Chile	0.48	0.09	0.07	0.01	50%	0.05
Argentina	0.24	0.19	0.00	0.01	60%	0.04
Ukraine	0.02	0.18	0.44	0.01	70%	0.03
Philippines	0.14	0.08	1.00	0.01	80%	0.02
Uruguay	0.41	0.08	0.07	0.01	90%	0.01
Total				0.69	100%	0

To standardize the quantitative scale, the values of the political risk index, export capacity level, and inverse of transportation distance in the table are standardized by extreme values. Description of indicators: (1) political risk index: whether the political environment of major exporting countries is safe and stable; (2) export capacity level: the proportion of a country’s agricultural exports in total global exports; (3) inverse of shortest route: the inverse of the shortest distance of actual trade between two countries; (4) export security degree: the product of the first three to indicate the stability of exports; (5) supply country disruption ratio: supply disruptions to the top ten countries in order; (6) stability of China’s agricultural imports: a quantitative indicator indicates the stability of China’s agricultural imports.

## Data Availability

Data is contained within the article.

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
