# Peer review of "Trade for Food Security: The Stability of Global Agricultural Trade Networks"

_foods, 2023, doi:10.3390/foods12020271_

Round 1
Reviewer 1 Report
Dear Authors
Congratulations for your manuscript.
Please see attached my comments and suggestions.
I hope they will be useful.
Best
--------------------------------------
General Comments
The manuscript presents an interesting discussion about the stability of global agricultural trade networks.
The manuscript is well written, the methodology applied is correct and the research shows interesting results.
However, the discussion is quite superficial and the conclusion does not provide enough information to answer the research questions described in the introduction.
As the authors said in some passages, the stability of agricultural trade networks depends on a set of factors such as trade policies, national relation, logistics structures, economic development…. but these issues were not accordingly considered in the discussion section to improve the interpretation of the results. Therefore, the conclusion and contribution of the current version of the manuscript is limited.
More comments:
Lines 88 - 91: “Since the stability of agricultural trade is affected by multiple factors such as political risks, economic benefits, geographical distance, national relations, and cultural attributes, its corresponding research needs to conduct throughout the network.”
What is the main contribution of this research for this issue?
Lines 96 - 97: “In the introduction we outline the importance of ATN stability for global food security”.
This issue is not enough addressed. The authors must provide more evidences and examples to clarify the important of ATN for global food security.
Lines 109 - 112: Provide references.
Line 111: Why the set P encompass 228 countries?
Lines 114 – 115: The authors must provide more information to support their conclusion: “agricultural trade shows strong grouping characteristics”.
Please, adjust all mathematical notations along the sentences. They are not aligned.
Line 50: ANT? Please, provide a complete definition at the first time the acronym is used on the main text.
Lines 72 - 91: None information from Brazil, a key producer for relevant commodities like soybean, meat, cotton, coffe..... and a crucial player for meet the increasing demand for food globally?
Lines 96 - 98: Provide a more detailed structure of the manuscript.
Line 97: Applied method (ATN)?
Line 174: Equation 10?
Lines 172 - 178: Passages with similar meanings.
Line 180: How the “hub” node is determined?
The section 3.1 does not provide any results. It must be moved to introduction or discussion sections.
Line 210: How the authors explain the “changing structure of communities in the global ATN” ? I imagine that this changing process is crucial to understand the four stages outlined in the Figure 1.
Period 1 and 2 show the same years (1986 – 1997)
The authors should keep the colors over the 4 periods described in the Figure 1. It would facilitate comparisons.
Lines 262 -263: And the increasing income for developing countries?
Line 266: North Africa or Africa? Please, use the same label for the categories.
Figure 2: The authors should consider the same countries for each diagram. This would improve the comparative analysis.
How the authors explain the relations between the results provide in lines 289 – 290: “….Belgium maintained a relatively lower satisfaction….” and line 312- 314 “the highest node stability was 0.69 in Belgium….”?
Line 300: How the authors explain this result? It indicates low or high average node stability?
Line 324: This is an expected result? The authors must explain this result. Moreover, this result is bad or good for ATN?
Lines 350 - 352: This is an expected result for any country, right? Therefore, I imagine that it is not a particular vulnerability of China.
Lines 387 – 388: Please, use the same label to describe the categories (Southeast Asia-Central North America).
Lines 397 – 400: The authors should consider the Brazil role for agricultural supply sustainability over the next years. Several international governments agencies such as ONU and FAO consider Brazil as a key player for provide food for increasing demand worldwide.
Lines 405 – 406: How international trade network can support climate change-related challenges?
Lines 450 – 456: The authors indicate two very relevant issues for ATN nowadays: Russia-Ukraine war and Covid-19, but do not discuss the implications of these factors. They must be explored as example of supply shock and how the ANT deal with then and how both impacts the ATN stability. Moreover, the authors should discuss how key players at ATN, as China, reacted to those supply shock.
Conclusion section: The authors answered the questions proposed in the introduction section. However, the answers are not complete. For example:
1- The authors do not explain accordingly the reasons of changes on the structure of the global ATN.
2- The authors do not provide enough information to detail neither which changes are characterizing the stability of the global ATN nor how changes in key node affects the overall stability of the ATN.
3- The authors offer only superficial information about the potential threat to the global food supply security from disruption in major trade relations.
The conclusion section should be reorganized.
Author Response
Please see the attachment, all questions have been answered.

Reviewer 2 Report
The paper studies agricultural trade networks with network analysis. The paper is interesting and brings new perspectives for global trade. The introduction and background is good with sufficient details for non-experts. There are few issues that need to be resolved.
The notation needs to be improved. Some variables and constants are not aligned with the text (pp 4 and 5).
Terms such as aij are not defined. The values of kj and ki are given in what units? The weights of the edges are not clear to me how they were defined.
Author Response

(The authors gave the same response as above.)

Reviewer 3 Report
I appreciate the authors’ effort to analyse the agricultural trade to evaluate the stability of global agricultural trade networks. Still, some issues have to be improved or clarified.
Firstly, The brief description of chosen 228 countries (selection methodology, f.e. % share representation of countries from continents, etc.)
2. In Chapter 3.1.1 it would be useful to add time period in the chapter’s topic
3. line 247 add time period for the used data in the interpretations of development
4. Add an explanation of chosen countries in chapter 3.1.2
5. Based on the study analyses mentioned, the reason for stable European cooperation is the implementation of CAP
6. In the chapter introduction, the authors mention the impact of climate change on global agricultural production. It would be useful to mention it as well in the chapter discussion
Author Response

(The authors gave the same response as above.)

Round 2
Reviewer 1 Report
Dear Authors,
Congratulations for this new version of the manuscript.
Now, I imagine that it displays the qualities need to be published.
Best